# The New Di-Gold Metallotweezer Based on an Alkynylpyridine System

**DOI:** 10.3390/molecules27123699

**Published:** 2022-06-09

**Authors:** Susana Ibáñez

**Affiliations:** Institute of Advanced Materials (INAM), Centro de Innovación en Química Avanzada (ORFEO-CINQA), Universitat Jaume I, Av. Vicente Sos Baynat s/n, 12071 Castellón, Spain; maella@uji.es

**Keywords:** metallotweezer, gold(I), host-guest chemistry, polycyclic aromatic hydrocarbons (PAHs)

## Abstract

We developed a simple method to prepare one gold-based metallotweezer with two planar Au-pyrene-NHC arms bound by a 2,6-bis(3-ethynyl-5-tert-butylphenyl)pyridine unit. This metallotweezer is able to bind a series of polycyclic aromatic hydrocarbons through the π-stacking interactions between the polyaromatic guests and the pyrene moieties of the NHC ligands. The metallotweezer was also used as a host for the encapsulation of planar metal complexes, such as the Au(III) complex [Au(C^N^C)(C≡CC_6_H_4_-OCH_3_-*p*)], for which there is a large binding constant of 946 M^−1^.

## 1. Introduction

The modification of the properties experienced by guest molecules encapsulated in the cavities of supramolecular structures have attracted the interest of chemistry researchers during the last three decades [1,2]. The most efficient biologic receptors are enzymes, which are conformationally flexible so that they can adapt to respond to the shape of specific guest molecules [3,4]. Artificial hosts try to mimic such induced-fit conformational change as a strategy for maximizing the binding with guests [5,6,7,8]. However, most synthetic hosts are have structures with little flexibility, and thus the conformational changes observed are always smaller than those shown in biological receptors.

During the last few years, we reported a series of metallotweezers based on the use of two pyrene-bis-imidazolylidene-gold(I) arms bound with four different bis-alkynyl spacers (Figure 1) [9,10,11,12,13,14,15]. These tweezers benefit from the tendency of Au(I) complexes to show linear geometry, and from the ability of the pyrene moieties to establish π–π-stacking interactions. This, together with the metallophilic interactions that Au can establish with some third-row transition metals [16,17,18,19,20], make these supramolecular entities behave as effective receptors for planar organic molecules and some selected square planar transition metal complexes. In the course of our research, we found that the supramolecular properties of our metallotweezers were greatly influenced by the nature of the spacer connecting the two flat arms of the molecule. In particular, we observed that the tweezer with the anthracenyl spacer **A** had a great tendency to dimerize forming non-covalently bound self-aggregated duplex complexes [9,10,12]. In contrast, the complex connected with the xanthenyl spacer **B** served as a metalloligand for metals such as Cu^+^, Ag^+^, and Tl^+^ [11]. The complex with the carbazolyl spacer **C** was able to encapsulate planar aromatic guests by approaching its two polyaromatic arms, and thus showed an interesting example of a guest-induced-fit conformational arrangement [13,14]. Finally, the metallotweezer with the rigid dibenzoacridine spacer **D** allowed preparing a mechanically-interlocked dimer, which we named clippane, a term that we coined to refer to two-component MIMs formed by two entangled molecular tweezers [15]. Mechanically interlocked molecules (MIMs) are molecules that are held together because of their topologies, which favor the formation of the so-called mechanical bonds. A mechanical bond, as was defined by Stoddart, is an entanglement in space between two or more components, such that they cannot be separated without breaking or distorting covalent bonds between atoms [21].

As shown in Figure 1, all the linkers that we used for the preparation of our metallotweezers are rigid, and therefore they were used with the aim to lead to strong and selective binding to planar organic molecules. Flexible spacers, however, are more prone to adapting their conformation to maximize substrate binding and hence operate through an ‘induced-fit’ mechanism, although the binding affinities are normally reduced due to the energy cost and entropy loss associated with the distortions and reduction of conformational changes [22,23]. In this regard, during the last decade, Yam [24,25,26,27,28] and Wang [29,30,31,32,33], prepared a series of alkynylplatinum tweezers bound by a rather flexible diphenylpyridine spacer, whose binding affinities were found to be perturbed by π–π and metal–metal interactions, producing dramatic color changes with diverse applications, such as the amplification of chiroptical signals [33] and visible-light photocatalytic transformations [34].

With these precedents in mind, we thought that we could expand the family of our tweezers by using a flexible diphenyl-pyridine spacer and study its abilities to form host-guest complexes with planar organic and inorganic molecules. With all this in mind, we now report the preparation of a tweezer containing a di-gold metallotweezer that combines two pyrene-bis-imidazolylidene ligands with one 2,6-bis(3-ethynyl-5-tert-butylphenyl)pyridine linker.

## 2. Results and Discussion

### 2.1. Synthesis and Characterization

The metallotweezer **2** was obtained by following the synthetic procedure shown in Figure 2. The reaction between the pyrene-imidazolylidene-gold(I) complex **1 [35]** and 2,6-bis(3-ethynyl-5-tert-butylphenyl)pyridine [27] in refluxing methanol in the presence of NaOH afforded the dimetallic complex **2** in high yield (67%) (Figure 2). Complex **2** was characterized by means of NMR, IR and Uv-Vis spectroscopy, and mass spectrometry, and provided satisfactory elemental analysis. Both the ^1^H and ^13^C NMR spectra are consistent with the twofold symmetry of the molecule. The ^13^C NMR spectrum of **2** showed a signal at 193.06 ppm, which is characteristic of the Au-C_carbene_ carbon. The mass spectrum showed a peak at *m*/*z* 1717.8, which was assigned to [2+H]^+^. The infrared spectrum of **2** showed the absorbance of the C≡C stretching at 2099.14 cm^−1^. The electronic spectrum of **2** in CH_2_Cl_2_ showed absorptions in the 270–370 nm region, which were assigned to π–π* intraligand transitions of the pyrene and diphenyl-pyridine moieties of the molecule. The emission spectrum of the same molecule in CH_2_Cl_2_ showed a strong vibronically resolved band between 370–490 nm, coincident with the typical monomer emission band of pyrene.

When the reaction was carried out in the presence of one equivalent of a planar Au(III) complex **3** [36], the corresponding inclusion complex **3@2** was obtained in 78% yield. This large yield is explained by the templation effect produced by the addition of the planar Au(III) complex containing a CNC pincer ligand [36]. The mass spectrum (ESI-MS) of **3@2** showed a prominent peak at *m*/*z* 2274.9, which is due to [3@2+H]^+^, (Appendix A). The Diffusion Ordered NMR spectrum (DOSY) showed that all the resonances displayed the same diffusion coefficient, indicating that the molecule of [Au(C^N^C)(C≡CC_6_H_4_-OCH_3_-*p*)] was associated with the molecular tweezer **2** forming a single assembly with **3** (DOSY; Appendix A).

### 2.2. Molecular Recognition: Determination of Binding Affinities

In order to quantify the extent of the binding between **2** and the different guests as electron-rich, electron-poor, and the complex of Pt(II) or Au (III) with 2,6-diphenylpyridine (**3**) as ligands, we performed a series of ^1^H NMR titrations to determine the association constants for the complexes formed. These were performed in CDCl_3_, at room temperature, and at a constant concentration of **2** (1.0 mM). The ^1^H NMR titrations showed that, in all cases, the addition of the PAH guest induced the upfield shifting of the signals due to the protons of the pyrene units of **2**, thus evidencing that the formation of the inclusion complexes showed fast kinetics on the NMR timescale. As an example, Figure 1 shows the selected region of ^1^H NMR spectra resulting from the titration **2** of **3**. For this case it can observed the resonance due to the proton of the N-C*H*_2_ group was shifted by–0.03 ppm. In addition, all three signals due to the protons of the pyrene moiety of the receptor were modified, which is a clear indication that they interact with the guest through a π–π stacking event. Based on the changes observed, the constants were determined by global fitting analysis [37,38], by processing the data using a 1:1 stoichiometric model. The results indicated that the binding affinities of the electron-rich PAH guests were in the order pyrene < triphenylene < perylene < coronene, as can be observed from the values shown in Table 1. This order, together with the relative changes in the binding constants, are consistent with the trend observed when comparing the binding affinities of PAH guests with hosts with large portals and is a consequence of the more effective π–π-stacking overlap produced as the electron richness of the guest increases (see Figure 2) [39,40,41]. The effect of adding a hydrogen-bonding group to the PAH molecule has a positive effect in the resulting binding constant, as can be seen when comparing the values obtained for pyrene and 1-pyrenyl-methanol (entries 1 and 5) and perylene and 3-perylenyl-methanol (entries 3 and 6). In both cases the incorporation of the hydroxyl group to the periphery of the PAH guest produces increased in the association constant, very likely due to the stabilization produced by the hydrogen bonding interaction between the –OH group and the lone pair of the nitrogen at the pyridine linker. If we compare the binding affinities of the metallotweezer with those observed with the carbazole-connected metallotweezer [42], higher affinity is observed with the pyridine linker-based tweezer, **2**, except that the rigid tweezer based on the dibenzoacridine-connected metallotweezer [42] has constants of association which are much higher than those observed in the pyridine linker-based tweezer, **2**.

In the case of encapsulation of electron-poor molecules such as 2,4,7-trinitro-9-fluorenone (TNFLU) and N,N′-dimethyl-naphthalenetetracarboxydiimide (NTCDI) (entries 7 and 8) we observed lower association constant values. It is noteworthy that for the TNFLU ligand, no change in the chemical shift of its signals was observed. This result is totally different from the one observed in the other two clamps synthesized previously, where a large increase in the association constant was observed, about 9933 M^−1^ [42]. With regard to the encapsulation of metal complexes, some authors previously showed how metallotweezers containing cationic alkynylplatinum arms displayed large association constants with several neutral and anionic guest planar metal complexes, including [Pt(C^N^C)(CO)] and [Au(C^N^C)(C≡CC_6_H_4_-OCH_3_-*p*)]. By using their alkynylplatinum-containing tweezer, Yam and co-workers [27] observed that the association constant with **3** was found to be logK = 4.43 under the same experimental conditions where we obtained an association constant of 946 M^−1^ (logK = 2.98) for the encapsulation of the same metal complex with **2** (entries 10). The lower affinity found in our case is very likely due to the presence of the electron-rich pyrene moieties, compared with the electron-deficient planar cationic arms present in the Yam’s case, which are more prone to favor the electrostatic interaction with planar electron rich metal complexes. In addition, in the Yam’s case, the binding affinity between the host and the guest was strongly perturbed by Pt^…^M interactions, whereas metallophillic interactions seem not to play a role in the formation of the host-guest complex **3@2**.

### 2.3. Photophysical Characterization

We wanted to have an estimation of the charge transfer produced in the interaction between the electron-donating host and the different electron deficient guests upon host–guest formation. Unfortunately, even the highest association constants that we found were not large enough to produce significant amounts of the host–guest complexes at the concentrations used to perform Uv-Vis and emission spectra (10^−4^–10^−5^ M), and therefore we discarded these spectroscopic techniques for our studies.

The absorption spectra of the compounds show the characteristic bands of this type of system and in the case of the encapsulation of the planar Au(III) complex there was an increase in intensity (Appendix A). The spectra showed a featureless strong band centered at 260 nm, assigned to the intraligand (IL) π–π* transitions of the alkynyl ligands, and a series of bands between 280–370 nm due to the absorption of the polyaromatic spacer and the pyrene moieties. On the other hand, the emission spectra of **2** and **3@2** showed a strong luminescence featuring two vibronically resolved bands with peak maxima at 400 and 379 nm, which are coincident with the typical monomer emission bands of pyridine and pyrene, respectively, in related diacetylide di-gold(I) complexes [43] and pyrene-based NHC ligands [44,45]. However, in the complex **3@2** a lower-energy emission band was observed at 450–550 nm (Appendix A) in the electronic emission spectra. In view of the fact that both the host complex **2** and the guest **3** do not absorb in this region, the formation of a new emission band is probably derived from the host–guest interaction resulting from the intercalation of the guest molecule into the cavity **2**. 

## 3. Conclusions

In this work we showed the encapsulating properties of a di-gold metallotweezer toward a large series of planar guests. The receptor was benefited by the presence of a 2,6-bis(3-ethynyl-5-tert-butylphenyl)pyridine bis-alkynyl linker, which allowed the parallel orientation of the two pyridine-imidazolylidene-Au(I) panels at a distance of about 7 Å. When the reaction was carried out in the presence of the planar Au(III) complex, the reaction directed towards the inclusion complex **3@2**. The association constants were a few lower than those observed for the other spaces based on the carbazolyl and dibenzoacridine linker, due to the flexibility of the spacer, except for the Au(III) having similar association constants. The lower affinity found in our case is very likely due to the presence of the electron-rich pyrene moieties, compared with the electron-deficient planar cationic arms present in Yam’s case, which are more prone to favor the electrostatic interaction with planar electron-rich metal complexes.

## 4. Materials and Methods

### 4.1. General Procedures

The NHC-Au(I) complex **1** [35], 2,6-bis(3-ethynyl-5-tert-butylphenyl)pyridine (**A**) [27], N,N′-dimethyl-naphthalenetetracarboxy diimide (NTCDI) [46], [Pt(C^N^C)(CO)] [47], and [Au(C^N^C)(C≡CC_6_H_4_-OCH_3_-*p*)] (**3**) [36] were prepared according to literature methods. All other reagents were used as received from commercial suppliers. 

### 4.2. Physical Measurements

Infrared spectra (FTIR) were performed on a FT/IR-6200 (Jasco) spectrometer equipped with a Pro One ATR with a spectral window of 4000–400 cm^−1^. NMR spectra were recorded on a Bruker 400 MHz using CDCl_3_ as solvents. High Resolution Mass Spectra (HRMS) were recorded on a Q-TOF Premier mass spectrometer (Waters) with an electrospray source operating in the V-mode. Nitrogen was used as the drying and cone gas at flow rates of 300 and 30 Lh^−1^, respectively. The temperature of the source block was set to 120 °C, and the desolvation temperature was set to 150 °C. Capillary voltage of 3.5 kV was used in the positive scan mode and the cone voltage was adjusted typically to 20 V. Mass calibration was performed by using solutions of NaI in isopropanol/water (1:1) from *m*/*z* 50 to 3000. Elemental analyses were carried out on a TruSpec Micro Series. UV/Visible absorption spectra were recorded on a Varian Cary 300 BIO spectrophotometer using CH_2_Cl_2_ under ambient conditions. Emission spectra were recorded on a modular Horiba FluoroLog-3 spectrofluorometer employing degassed CH_2_Cl_2_.

### 4.3. Synthesis and Characterization

#### 4.3.1. Synthesis of **2**

NaOH (40.00 mg, 1.002 mmol) and 2,6-bis(3-ethynyl-5-*tert*-butylphenyl)pyridine (**A**, 28.01 mg, 0.072 mmol) were placed together in a round-bottom flask and dissolved in methanol (20 mL). This mixture was heated at reflux for 1 h. Then, complex **1** (100.00 mg, 0.143 mmol) was added as a solid and the resulting suspension was heated at reflux for 4 h. The mixture was evaporated to dryness and the solid residue was extracted with dichloromethane, and the solution was filtered through a short pad of Celite. Complex **2** was isolated as a white solid. Yield: 82.45 mg (67%). IR (ATR): ʋ (C≡C): 2099.14 cm^−1^. HRMS ESI-TOF-MS (positive mode): 1717.8 [2+H]^+^ (Appendix A). Anal. Calcd. for C_95_H_111_N_5_Au_2_: C, 66.46; H, 6.52; N, 4.08. Found: C, 66.50; H, 6.51; N, 4.06. ^1^H NMR (400 MHz, CDCl_3_): *δ* 8.66 (d, 4H, C*H*_pyr_), 8.40 (t, ^4^*J*_H-H_ = 4.0 Hz, 2H, C*H*_phenyl_), 8.25 (d, 4H, C*H*_pyr_), 8.09 (s, 4H, C*H*_pyr_), 8.02 (t, ^4^*J*_H-H_ = 4.0 Hz, 2H, C*H*_phenyl_), 7.79 (dd, ^3^*J*_H-H_ = 8.0 Hz, ^3^*J*_H-H_ = 8.0 Hz, 1H, C*H*_pyridine_), 7.75–7.70 (m, 4H, C*H*_pyridine_+C*H*_phenyl_), 5.27 (t, ^3^*J*_H-H_ = 8 Hz, 8H, NC*H*_2_CH_2_CH_2_CH_3_), 2.22–2.10 (m, 8H, NCH_2_C*H*_2_CH_2_CH_3_), 1.79–1.66 (m, 8H, NCH_2_CH_2_C*H*_2_CH_3_), 1.62 (s, 36H, C(C*H*_3_)_3_), 1.44 (s, 18H, C(C*H*_3_)_3 phenyl_), 1.05 (t, ^3^*J*_H-H_ = 8 Hz, 12H, NCH_2_CH_2_CH_2_C*H*_3_) (Appendix A). ^13^C{^1^H} NMR (100 MHz, CDCl_3_): *δ* 193.06 (Au-*C*_carbene_), 156.83 (*C*_q phenyl_), 151.28 (*C*_q phenyl_), 149.28 (*C*_q pyr_), 138.88 (*C*_q pyr_), 137.48 (*C*H _pyridine_), 132.00 (*C*_q pyr_), 130.71 (*C*H _phenyl_), 128.48 (*C*H_pyr_), 128.04 (*C*H _phenyl_), 127.97 (*C*_q pyr_), 127.45 (*C*_q pyridine_), 125.41 (*C*_q pyr_), 123.01 (*C*H_pyr_), 122.99 (*C*H _phenyl_), 121.75 (*C*_q acetylide_), 120.92 (*C*_q acetylide_), 118.59 (*C*H _pyridine_), 116.82 (*C*H_pyr_), 106.23 (*C*_q phenyl_), 52.36 (N*C*H_2_CH_2_CH_2_CH_3_), 35.62 (*C*(CH_3_)_3_), 35.06 (*C*(CH_3_)_3 phenyl_), 32.81 (NCH_2_*C*H_2_CH_2_CH_3_), 32.00 (C(*C*H_3_)_3_), 31,51 (C(*C*H_3_)_3 phenyl_), 20.42 (NCH_2_CH_2_*C*H_2_CH_3_), 14.16 (NCH_2_CH_2_CH_2_*C*H_3_) (Appendix A).

#### 4.3.2. Synthesis of **3**@**2**

NaOH (40.00 mg, 1.002 mmol) and 2,6-bis(3-ethynyl-5-*tert*-butylphenyl)pyridine (**A**, 28.01 mg, 0.072 mmol) were placed together in a round-bottom flask and dissolved in methanol (20 mL). This mixture was heated at reflux for 1 h. Then, complex **1** (100.00 mg, 0.143 mmol) and **3** (40.13 mg, 0.072 mmol) were added as solid and the resulting suspension was heated at reflux for 4 h. The mixture was evaporated to dryness and the solid residue was extracted with dichloromethane, and the solution was filtered through a short pad of Celite. Complex **3@2** was isolated as a white solid. Yield: 128.20 mg (78%). IR (ATR): ʋ (C≡C): 2145.42 and 2105.89 cm^−1^. HRMS ESI-TOF-MS (positive mode): 2274.9 [3@2+H]^+^ (Appendix A). Anal. Calcd. for C_121_H_129_N_6_OAu_3_: C, 63.90; H, 5.72; N, 3.70. Found: C, 63.86; H, 5.74; N, 3.70. ^1^H NMR (400 MHz, CDCl_3_): *δ* 8.65 (s, 4H, C*H*_pyr_), 8.39 (s, 2H, C*H*_phenyl_), 8.23 (d, 4H, C*H*_pyr_), 8.07 (s, 4H, C*H*_pyr_), 8.04 (d, ^3^*J*_H-H_ = 8.0 Hz, 2H, C*H*_Au(III)_), 8.02 (s, 2H, C*H*_phenyl_), 7.80 (dd, ^3^*J*_H-H_ = 8.0 Hz, ^3^*J*_H-H_ = 8.0 Hz, 1H, C*H*_pyridine_), 7.76–7.69 (m, 5H, C*H*_pyridine_+C*H*_phenyl_+C*H*_Au(III)_), 7.36 (dd, ^3^*J*_H-H_ = 15 Hz, ^3^*J*_H-H_ = 9.0 Hz, 4H, C*H*_Au(III)_), 7.40 (t, ^3^*J*_H-H_ = 15 Hz, ^3^*J*_H-H_ = 9.0 Hz, 4H, C*H*_Au(III)_), 7.24 (t, ^3^*J*_H-H_ = 8.0 Hz, ^3^*J*_H-H_ = 6.0 Hz, 2H, C*H*_Au(III)_), 6.87 (d, ^3^*J*_H-H_ = 9 Hz, 2H, C*H*_Au(III)_), 5.25 (t, ^3^*J*_H-H_ = 8 Hz, 8H, NC*H*_2_CH_2_CH_2_CH_3_), 3.84 (s, 3H, OC*H_3_*_Au(III)_), 2.25–2.08 (m, 8H, NCH_2_C*H*_2_CH_2_CH_3_), 1.79–1.66 (m, 8H, NCH_2_CH_2_C*H*_2_CH_3_), 1.62 (s, 36H, C(C*H*_3_)_3_), 1.44 (s, 18H, C(C*H*_3_)_3 phenyl_), 1.04 (t, ^3^*J*_H-H_ = 8 Hz, 12H, NCH_2_CH_2_CH_2_C*H*_3_) (Appendix A). ^13^C{^1^H} NMR (100 MHz, CDCl_3_): *δ* 195.95 (Au-*C*_carbene_), 167.06 (*C*_q Au(III)_), 165.03 (*C*_q Au(III)_), 156.80 (*C*_q phenyl_), 151.27 (*C*_q phenyl_), 149.23 (*C*_q_), 149.09 (*C*_q_), 142.12 (*C*_q_), 138.86 (*C*_q_), 137.50 (*C*H _pyridine_), 136.80 (*C*H_Au(III)_), 133.28 (*C*H_Au(III)_), 132.13 (*C*H _phenyl_), 131.95 (*C*H_Au(III)_), 130.72 (*C*H _phenyl_), 128.45 (*C*H_pyr_), 128.00 (*C*H_Au(III)_), 129.75 (*C*_q_), 126.81 (*C*H_Au(III)_), 125.38 (C_q_), 125.16 (*C*H_Au(III)_), 122.98 (*C*H_pyr_), 122.89 (*C*H _phenyl_), 121.71 (*C*_q acetylide_), 120.89 (*C*_q acetylide_), 119.07 (*C*_q acetylideAu(III)_), 118.59 (*C*H _pyridine_), 118.57 (*C*_q acetylideAu(III)_), 116.82 (*C*H_pyr_), 116.70 (*C*H_Au(III)_), 113.82 (*C*H_Au(III)_), 106.21 (*C*_q phenyl_), 55.46 (O*C*H_3_), 52.33 (N*C*H_2_CH_2_CH_2_CH_3_), 35.61 (*C*(CH_3_)_3_), 35.05 (*C*(CH_3_)_3 phenyl_), 32.80 (NCH_2_*C*H_2_CH_2_CH_3_), 32.00 (C(*C*H_3_)_3_), 31.50 (C(*C*H_3_)_3 phenyl_), 20.42 (NCH_2_CH_2_*C*H_2_CH_3_), 14.17 (NCH_2_CH_2_CH_2_*C*H_3_) (Appendix A).

## Data Availability

The data presented in this study are available on request from the corresponding author.

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
