# Peer review of "The New Di-Gold Metallotweezer Based on an Alkynylpyridine System"

_molecules, 2022, doi:10.3390/molecules27123699_

Round 1
Reviewer 1 Report
see attached

Author Response
Thanks for the comments and for the suggestions. The responses are responded to point by point in the pdf document.

Reviewer 2 Report
This manuscript is the review described about digold complexes like tweezer with a flexible diphenyl-pyridine spacer and formation host-guest complexes with planar organic and inorganic molecule.The system is attractive and obtained products are also characterized definitely.
The manuscript should be published but before that, it would be grateful if you answer and/or reflect the feedbacks in the manuscript if necessary.
(1) Where do you guess the guest molecule is positioned on encapsulation? In line 102, you said, “all three signals due to the protons of the pyrene moiety of the receptor are modification, which is a clear indication that they interact with the guest through a π- π stacking event.” Maybe you expect the structure as a guest molecule is sandwiched between pyrenes of the receptor, I think. However, alkynyl metal complexes proposed in manuscript have only poor conjugated system and thus are expected to be encapsulated by metallophilic interaction rather than π- π stacking and are positioned near gold moieties in receptor. If so, the signals due to the protons of the pyrene moiety are hardly modified, I wonder. Do you have any ideas?
(2)  In photophysical properties, you said, “in the complex 3@2 is observed another band a lower energy absorption band at 450-550 nm (Figure S6) in the electronic absorption spectra.” First, S6 shows emission spectra not absorption. Does the description in line 171-175 explain luminescent properties not absorption properties? I confused.
On the presumption that it is the explanation about the emission, you said, “the formation of a new absorption band is probably derived from the host-guest interaction resulting from the intercalation of the guest molecule into the cavity 2”. It is sure that occurring of new emission band suggested “the presence” of the guest complex molecules, however, it does not necessarily show “the intercalation” of the guest molecules. If intercalation occurred, luminescent properties may be different from free metal complexes, e.g., energy shifts, occurring of mixed-metal CT band and emission extinction by energy transfer. To prove the intercalation, comparison of luminescent properties of encapsulated complexes with that of free guest molecules is necessary, I think.
Author Response
Thanks for the comments and for suggestions. About your question of where the guests will be placed when encapsulated. The guests are encapsulated in the middle of the cavity by p-p interactions. Their host-guest interactions have been by NMR spectroscopy. For this metallotweezer, small variations have been observed in the chemical shift of the CH group signals from the pyrene groups and n-Bu substituents, so they may be in the cavity but shifted towards the gold metal centers and the alkynyl groups. In fact, we have demonstrated in our previous studies (Chem. Eur. J. 2021, 27, 9661-9665, Acc. Chem. Res. 2020, 53, 1401-1413) where we were able to obtain the X-ray structure of some host-guest adduct in which the guests are placed in an offset position towards the interior of the cavity.
With respect to your second question, first I have modified the confusion I generated in the band appearing in the emission spectrum at 450-550nm. This point has been addressed (see the highlighted version of the corrected manuscript).
In the supplementary information is shown in Figure S6 the emission spectrum of compound 3, the free host. It was measured under the same conditions as compound 2 as of 3@2. And as can be seen the lowest energy emission band is observed at 450-550 nm in the electronic emission spectrum. Considering that both the host complex 2 and the guest 3 do not absorb in this region, the formation of a new emission band is probably derived from the host-guest interaction. In fact, the professor Vivian Yam in her studies (Chem. Sci. 2012, 3,1185) demonstration that the origin of the host-guest interaction should be related to the metal-metal, p-p interactions as the steric factor. In the Yam’s case, the Uv-vis titration study of 3 show a decrease in the MLCT/LLCT absorption with no obvious grown of a lower-energy absorption. Emission spectral changes upon addition the Au(III) complex to host also showed quenching of the 3MLCT emission, while careful comparison of the normalized spectra indicated that there was a very small increase in the low-energy emission resulted from the host-guest interaction.
Reviewer 3 Report
Susana Ibanez reports on a simple method to prepare one gold-based metallo tweezers with two planar Au-pyrene-NHC arms bound by a 2,6-bis(3-ethynyl-5-tert-butylphenyl)pyridine unit. This metallo tweezer is able to bind a series of polycyclic aromatic hydrocarbons through π-stacking interactions between the polyaromatic guests and the pyrene moieties of the NHC ligands. The metallo tweezer was also used as host for the encapsulation of planar metal complexes, such as the Au(III) complex [Au(C^N^C)(C≡CC6H4-OCH3-p)], for which a large binding constant of 946 M‒1.
Evaluation:
In continuation of her work on metallo supramolecular interaction, preferably of gold complexes, the author presents a short report on a new Au-based metallo tweezer, which was inspired by previous work by Vivian Yam et al. The modification is small, but the author describes the previous system in detail and draws interesting comparisons. Moreover, Ibanez reports on detailed experiments of binding polyaromatic hydrocarbons through these tweezers. The work has been competently carried out, the data looks solid and the conclusions are backed from the data.
This is a very interesting contribution and the communication format is justified from the scope of this work.
I can recommend acceptance of the paper after minor revision.
Points for the revision:
- The English should be carefully revised. It is not really bad, but a nice paper deserves a good English. So, the author should look for help of an experienced English writer and revise the text.
- Title; ... on an alkynylpyridine system
- In the abstract it should read:... to bind a series of polycyclic aromatic hydrocarbons through π-stacking
- Abstract and everywhere: complex [Au(C^N^C)(....-p)], for which ...
- Page 3, line 87, I suggest to rewrite: ... (DOSY, Figure S28) showed that ... indicating that the complex [Au(C^N^C)(....-p)] (3) is associated ... forming the assembly 3@2 (Scheme 2)
- Complex parentheses also on page 4, lines 131 and 132. Please check the entire manuscript.
- Page 4, lines 128, 135. Please check all “‒1” and make sure to use the long hyphen (it is much better visible)
- Page 4, line 140, the “...” should be in the middle of the line
- Figure 12, please enlarge all labels by at least x3, they are almost unreadable
- Wrong or missing journal abbreviations: Ref 1, 7, 8, 33, 44, please correct
- typos in Ref. 19, 26, 35, 36, please correct
Author Response
Thanks for the comments and for the suggestions. All the minor points have been addressed (see highlighted version of the corrected manuscript).
Round 2
Reviewer 1 Report
Thank you for the corrections. However, I still do not understand how such tiny NMR shifts can give such big binding constants (even if they are relatively small compared to other metallotweezers).
Again, I point out my previous comment : "All the more so that, if one compares the shifts with those reported by the same author in the recent 2022 paper in Inorganic Chemistry (ref 41), one finds larger chemical shifts (by an order of magnitude) for similar binding constants (e.g. compare binding of 1-pyrenyl-methanol in the IC paper giving k = 179 and with shifts for NCH2 ranging from 5.21 to 4.41 ppm and the present work with binding of 3-perylenyl-methanol giving k = 169 and with shifts for NCH2 ranging only between 5.27 and 5.21."
This simply makes no sense to me. Barring a satisfactory explanation of this phenomenon, I am afraid that I must recommend to reject this manuscript.
Author Response
Thank you for your comments. I have answered your questions in the attached pdf. I hope you will find my answers helpful.
